# Reconfigurable and responsive droplet-based compound micro-lenses

Sara Nagelberg[1], Lauren D. Zarzar[2,3], Natalie Nicolas[1], Kaushikaram Subramanian[4], Julia A. Kalow[2,5], Vishnu Sresht[6], Daniel Blankschtein[6], George Barbastathis[1], Moritz Kreysing[4], Timothy M. Swager[2] & Mathias Kolle[1]

Micro-scale optical components play a crucial role in imaging and display technology, biosensing, beam shaping, optical switching, wavefront-analysis, and device miniaturization. Herein, we demonstrate liquid compound micro-lenses with dynamically tunable focal lengths. We employ bi-phase emulsion droplets fabricated from immiscible hydrocarbon and fluorocarbon liquids to form responsive micro-lenses that can be reconfigured to focus or scatter light, form real or virtual images, and display variable focal lengths. Experimental demonstrations of dynamic refractive control are complemented by theoretical analysis and wave-optical modelling. Additionally, we provide evidence of the micro-lenses' functionality for two potential applications—integral micro-scale imaging devices and light field display technology—thereby demonstrating both the fundamental characteristics and the promising opportunities for fluid-based dynamic refractive micro-scale compound lenses.

[1] Department of Mechanical Engineering, Massachusetts Institute of Technology, 77 Massachusetts Avenue, Cambridge, Massachusetts 02139, USA. [2] Department of Chemistry and Institute for Soldier Nanotechnologies, Massachusetts Institute of Technology, 77 Massachusetts Avenue, Cambridge, Massachusetts 02139, USA. [3] Department of Materials Science and Engineering and Department of Chemistry, The Pennsylvania State University, 427 Steidle Building, University Park, Pennsylvania 16802, USA. [4] Max Planck Institute of Molecular Cell Biology and Genetics, Pfotenhauerstr. 108, 01307 Dresden, Germany. [5] Department of Chemistry, Northwestern University, 2145 Sheridan Road, Evanston, Illinois 60208, USA. [6] Department of Chemical Engineering, Massachusetts Institute of Technology, 77 Massachusetts Avenue, Cambridge, Massachusetts 02139, USA. Correspondence and requests for materials should be addressed to M.K. (email: mkolle@mit.edu).

Micrometre-scale optical elements have contributed significantly to the miniaturization of devices and instrumentation. Static micrometre-sized lenses have utility in integral imaging and 3D displays[1–3], synthetic aperture imaging[4], endoscopes[5], plenoptic cameras[6] and solar concentrators[7]. Dynamically switchable reflective micro-optics based on digital micro-mirror displays[8] and continuously reconfigurable absorptive pixel technology enabled by liquid crystal displays[9] have greatly enhanced the versatility of micro-optics. These methods have enabled transformative advances in optical technology ranging from high-resolution displays to structured illumination microscopy, holographic optical tweezers and wavefront-shapers[10,11].

Morphological reconfiguration of micro-scale refractive components to enhance optical performance has recently been observed in the context of biological vision systems: rod photoreceptor cell nuclei in the retina of nocturnal mammals specifically adapt a bi-phase refractive index distribution to act as collecting lenses. These biological micro-lenses channel light towards the rods' light sensing segments, thereby increasing the signal to noise ratio of retinal transmitted images[12,13]. Similarly, man-made, refractive micro-elements with reconfigurable morphology, enabling in-situ tunable optical properties are poised to complement and extend the capabilities of present micro-optical devices[14–17]. In particular, tunable micro-lens designs employed as responsive in-line, phase-modulating, intensity-preserving components will extend the light manipulation capabilities of optical systems[10].

To this end, optofluidic devices using dynamic fluid lens materials represent an ideal platform to create versatile, reconfigurable, refractive optical components[17,18]. Droplets smaller than the capillary length, wherein surface tension is the dominant force, create curved interfaces between fluid volumes[19] and display intrinsic lensing behaviour. In addition, liquids have minimal surface roughness on the order of nanometres, even if the interfacial tension is very low[20,21]. Dynamic lensing materials based on hydrogels and liquids can be reshaped through various external stimuli after the optical element is formed, which is ideal for adaptive optics, imaging devices or sensors[14,22–24]. For example, reconfigurable liquid lenses have been demonstrated by taking advantage of electro-wetting[25–32] and the integration of microfluidics with MEMS technology[16]. Adjustable focal length lenses were realized using microfluidics, by varying the amount of liquid in a reservoir behind an elastic membrane[22,33–37]. Alternatively, the controlled flow of liquids within microfluidic channels can be used to create micro-lenses with variable focus[38,39]. Micrometre-sized solid-liquid doublet lenses that allow for minimization of optical aberrations have also been fabricated[33]. Tunable fluid micro-lenses, as individual components or arranged in arrays, have found applications in miniaturized optical components with variable working distances and optics-based biosensing devices[40,41]. Incorporation of dyes into the liquids allows for droplets that serve as both lenses and optical filters[42]. In particular, the incorporation of laser dyes in micro-fluidic droplets enables lasing-based sensing approaches with high sensitivity and throughput[43].

We present herein the optical characteristics of a new generation of fluidic tunable compound micro-lenses. These compound micro-lenses are composed of hydrocarbon and fluorocarbon liquids that form stable bi-phase emulsion droplets in aqueous media[44]. The choice of constituent liquids can dramatically impact the optical properties. In this initial study, we focus on combinations of transparent fluids Fluorinert FC-770 ($n_{FC}$=1.27) with heptane ($n_{HP}$=1.387), or hexane ($n_{HX}$=1.375). The refractive index of the hydrocarbon constituent is higher than the refractive index of water ($n_W$=1.33), while the fluorinated component has a refractive index lower than that of water. The refractive index contrast at each material interface as well as the curvature of each interface contributes to the focusing power of a refractive optical element (see lens maker's equation[45]). Therefore, we anticipated that these fluid combinations could allow for a wide tuning range of the emulsion lenses' focal length, thereby enabling switching between converging or diverging lens geometries. The complex droplet lenses can be easily fabricated on a large scale using a temperature-induced phase separation technique appropriate for combinations of liquids having a relatively low upper critical solution temperature[46]. Most importantly, such complex droplets can be dynamically reconfigured between double emulsion and Janus (two-sided) morphologies through application of external stimuli, which makes these droplets very promising as highly tunable compound lenses. We demonstrate the adjustability in focal length of the lenses as well as their microscopic and macroscopic light manipulation characteristics.

## Results

**Modelling of emulsion droplets as tunable lenses.** For these particular emulsions, the curvature of the internal interface formed between the immiscible phases can be adjusted using surfactants that modify the relative interfacial tensions between the droplet phases and water. Surfactant-mediated modification of interfacial tensions results in a variation of the contact angles at the triple-phase contact line. This determines the radius of curvature of the lenses' internal interface, which in turn affects the optical properties of the droplets (Fig. 1a,b). To demonstrate how the controlled, dynamic variation of the complex droplets' geometry could induce a tunable interaction with light, a ray-tracing algorithm was implemented in MATLAB. The overall droplet shape was assumed to be spherical, which is an appropriate approximation when the interfacial tension between the droplet phases is much smaller than the interfacial tensions between the droplet constituents and the aqueous medium (Fig. 1a) (ref. 47). This is the case for working temperatures close to the critical temperature of the internal fluids. For the droplet diameters on the order of 100 µm discussed here, the internal interface can be considered to be spherical, since the ratio of gravitational to surface tension forces is small (see discussion in Methods). Under these assumptions, the distance $l$ of the interface from the centre of the overall drop is given by

$$(R_d - l)^2 \left(l^2 + 4R_i R_d + 2R_d l - 3R_i l - 3R_d^2\right) + \frac{16 R_d (R_i - l)}{1 + v_r} = 0,$$
$$(1)$$

where $R_d$ is the radius of the drop, $R_i$ is the internal radius of curvature and $v_r$ is the volume ratio of the internal droplet phase to the outer droplet phase. The internal curvature is set by the balance of interfacial tensions at the triple-phase contact line[47] given by

$$\frac{\gamma_H - \gamma_F}{\gamma_{HF}} = \left(R_d^2 + 2R_i l - l^2\right)/(2R_i R_d). \quad (2)$$

The derivation of equations (1) and (2) can be found in Supplementary Note 1.

When the optical axis is aligned with the droplets' symmetry axis, the optical system is axisymmetric and can be modelled in two dimensions. The droplets' symmetry axis aligns with the gravitational field due to the difference in density between the light hydrocarbon phase and the dense fluorocarbon phase. We exploit this alignment in our theoretical and experimental

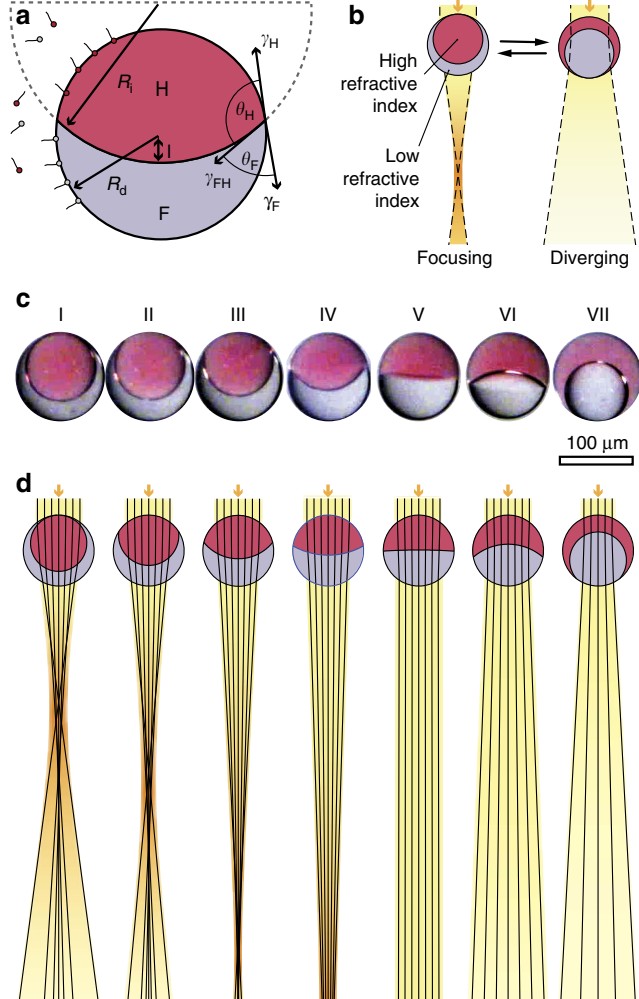

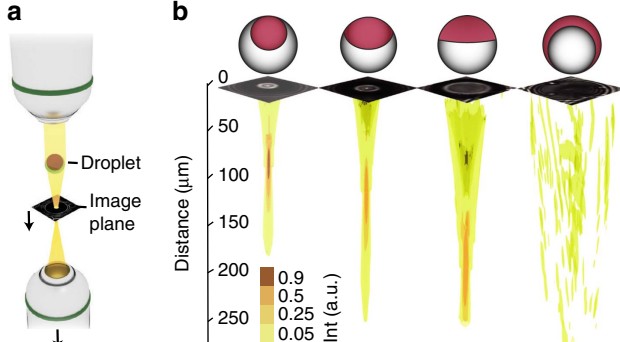

**Figure 1 | Concept and modelling of microfluidic emulsion-based compound lenses.** (**a**) Geometry of a bi-phase emulsion droplet. The internal curvature $R_i$ is determined by the interfacial tensions between hydrocarbon and water $\gamma_H$, fluorocarbon and water $\gamma_F$, and hydro- and fluorocarbon $\gamma_{FH}$. The distance from the centre of the drop to the internal interface $l$ is constrained by the volume ratio (equation (1)). (**b**) Double emulsions are expected to switch between focusing and diverging geometries. (**c**) Side view optical micrographs of droplets composed of FC-770 (grey) and heptane (red) with varying internal interface curvature. The red colour of the heptane phase results from the incorporation of the dye Sudan Red 7B. (**d**) Corresponding ray-tracing simulations showing the propagation of light rays through the droplets.

study of the droplets' optical characteristics. The ray-tracing calculations predict that the double emulsion droplets with a high refractive index core phase and a lower refractive index shell phase can focus light, while an inversion of this droplet geometry results in diverging lenses (Fig. 1c,d). By adjusting the droplets' internal interface curvature, each droplet can be tuned between a converging lens with varying positive optical power and a diverging lens with varying negative optical power. This optical behaviour is similar to the characteristic differences observed for the retina cell nuclei of nocturnal and diurnal mammals. Photoreceptor nuclei of nocturnal mammals concentrate on the highly refractive heterochromatin in the nuclear centre, while diurnal mammalian retina cells locate heterochromatin material towards their periphery[12]. The nucleus geometry with the higher refractive index material concentrated at the nucleus

**Figure 2 | Variable focusing.** (**a**) Schematic of the setup used to record the light field behind the droplets. The bottom objective scans through the z-direction. (**b**) Iso-surfaces of the reconstructed light fields behind the droplets for different internal droplet morphologies. Representative image data sets captured immediately behind the droplets show slices of the scan, from which the light field was reconstructed. Droplet sketches are offset upwards by 10 μm to not obstruct slice images.

centre focuses light, while an inversion of this nucleic geometry leads to strong light scattering, similar to the double emulsions.

**3D focus scans behind droplets with varying morphology.** Based on the ray-tracing predictions, we expected that altering the droplet morphology would induce a change in the droplet's focal length, which we aimed to demonstrate experimentally. In practice, the interfacial tensions that determine the droplet morphology can be tailored by controlling the concentration and ratio of surfactant species added to the aqueous phase. In our experiments, we used a combination of both a hydrocarbon-stabilizing surfactant, such as sodium dodecyl sulfate (SDS), and a fluorocarbon-stabilizing surfactant, such as Zonyl FS-300 or Capstone FS-30, as detailed in an earlier publication[44]. In order to map the light field behind heptane-FC-770 droplets, the droplets were illuminated with a collimated beam of quasi-monochromatic light of a 540 nm wavelength, and the light field in the volume behind the droplets was recorded by scanning the image plane of an inverted microscope (Fig. 2a). Variation in the concentrations of SDS and Capstone FS-30 in the surfactant mixture added to the aqueous phase allowed us to alter the droplets' internal interface curvature resulting in a variation of their focal length (Fig. 2b).

**Quantification of the droplet lenses' optical characteristics.** The native function of a lens is to form an image. In order to evaluate the image formation capabilities of the droplets, including experimentally quantifying their optical power, we placed an object in front of the droplets and used them to project an image at varying distances (Fig. 3a). Specifically, a grid pattern was projected in the aqueous medium above the droplets. The image of the object formed by the droplets was recorded using an inverted microscope. By varying the concentrations of SDS and Zonyl surfactant in the aqueous medium surrounding the droplets, the internal interface shape could be adjusted. By projecting the image of an object through the lenses and by measuring object-to-lens and lens-to-image distances (Fig. 3a–d), the micro-lenses' effective focal length was quantified (Fig. 3e; see Supplementary Note 2 for details regarding the derivation of the effective focal length).

The internal interface curvature of the droplets was determined by fitting a circle to the interface shape observed in side-view micrographs (Fig. 3e, inset), taking into account refraction due to

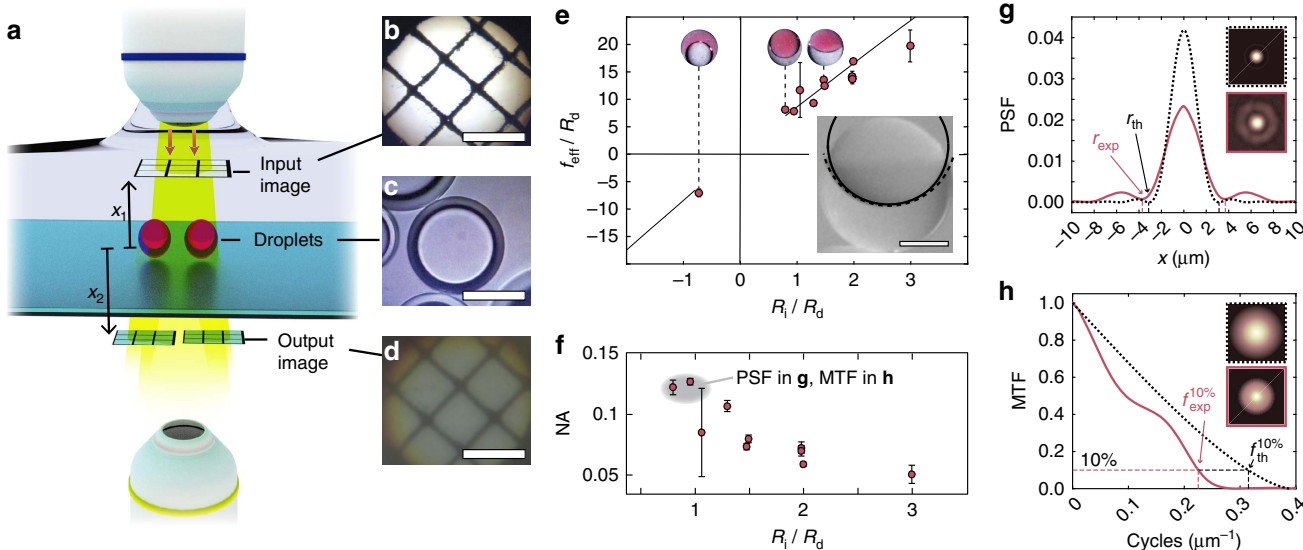

**Figure 3 | Characterization of the fluidic lenses' optical properties.** (**a**) Schematic of the optical setup used for focal length and image forming analysis. A grid image is projected in front of the droplets to serve as the object for the micro-lenses. The image formed by the droplets is recorded using a ×10 objective. (**b**) Image of a grid projected above the droplets. (**c**) A droplet viewed from above. (**d**) Image of the grid shown in **b** projected by the droplet displayed in **c**. Scale bars = 100 µm (**b–d**). (**e**) Effective focal length as a function of internal radius of curvature $R_i$, normalized by the droplet diameter $R_d$. Error bars represent uncertainty in fit parameter $f$ and uncertainty in measurement of internal curvature. The solid black line shows the expected focal length of the system given by the ray transfer matrix method. The bottom right inset shows a side view of a droplet with the same surfactant concentration as the droplet shown in **c**. The fit to the interface is shown with a solid line and the actual curvature is shown with a dashed line. Scale bar = 100 µm. (**f**) Numerical aperture NA as a function of internal radius of curvature $R_i$, given by $NA(R_i) = n \sin(\tan^{-1}(R_d/f(R_i)))$. Here, the refractive index is $n = 1$ (the image is formed in air). Error bars represent measurement uncertainty propagated from the uncertainty shown by the error bars in **e**. The grey-shaded area signifies the configuration of droplets for which the point spread function and the modulation transfer function are shown in **g,h**. (**g**) Point spread function estimate (PSF) of droplets with a numerical aperture NA = 0.12 for red light (pink line). The theoretical PSF for a diffraction-limited lens with identical NA is shown as a dashed black line. The area under the curves is normalized to unity. Insets show the theoretical 2D PSF (black dashed frame) and the experimentally determined PSF (pink frame). The experimentally determined two-point resolution limit amounts to $r_{exp} = 3.7$ µm. (**h**) Modulation transfer function (MTF) for the same droplets. The cut-off frequency above which the image contrast is less than 10% amounts to $f_{exp}^{10\%} = 0.22$ cycles per µm. Insets show the theoretical 2D MTF (black dashed frame) and the MTF determined from the experimentally obtained PSF (pink frame).

the outer droplet phase (see Methods). Knowing this curvature, we could compare the expected effective focal length acquired using the paraxial approximation and the ray transfer matrix calculations with the experimentally determined effective focal length (Fig. 3e). We found that FC-770-heptane droplets formed with volume ratio 1:1 can vary in focal length from 3.5 times the diameter of the droplet to infinity, and can switch between positive and negative focal lengths. For example, a double emulsion droplet of 100 µm diameter, with heptane as the core phase and the fluorocarbon FC-770 as the shell phase, has a focal length of 350 µm and acts as a converging lens. While we restricted the experiments presented here to lenses with constant volume ratio of 1:1, it is worth mentioning that a variation in volume ratio results in a change in radius of curvature of the internal interface and consequently in changes of the lenses' focal length provided that the triple phase contact angles are kept constant. Ray-tracing results of droplets with constant contact angle and varying volume ratio can be seen in Supplementary Fig. 3. This additional degree of freedom suggests interesting future perspectives for multi-fluid lens optical systems, especially in terms of higher order aberration correction.

The configuration of the droplet in Fig. 1c-V is a special case where the droplets have an effective focal length of infinity. For the FC-770-heptane emulsions, this occurs when the interface is nearly flat; the refraction at the water–heptane interface is effectively cancelled by a compensating refraction at the

FC-770–water interface. As expected, the lenses' numerical aperture, given by $NA = n \sin(\tan^{-1}(D/2f))$, decreases with increasing focal length. Here $n = 1$, since the image was formed in air beneath the droplet lenses, which were positioned on top of a glass coverslip.

To estimate the droplet lenses' optical quality, we determined two important metrics used in the design of lenses: First, the two-point resolution criterion postulated by Rayleigh in 1896 provides a measure for the minimum distance between two object points for which these two points can still be distinguished unambiguously in the image projected by a lens. Second, the Abbe diffraction limit defines the maximum spatial frequency of a sinusoidally varying intensity pattern that can be resolved with sufficient contrast by the lens. We apply the standard definition of the Rayleigh two-point resolution criterion, which consists of determining the distance from the centre of the lenses' point spread function (PSF) to its first minimum. With this definition, the theoretically achievable resolution $r_{th}$ of a diffraction-limited lens is given by $r_{th} = 1.22 \cdot (\lambda/2NA)$. We experimentally estimated the PSF of droplets by imaging the focus formed by droplets that were illuminated with collimated light. We used a white light source but only exploited the image information of the camera's red channel (with maximum quantum efficiency at 620 nm). The experiment was performed with droplets with a highly curved internal interface, assuming that their shorter focal length and higher numerical aperture NA ≈ 0.12 would result in the best obtainable

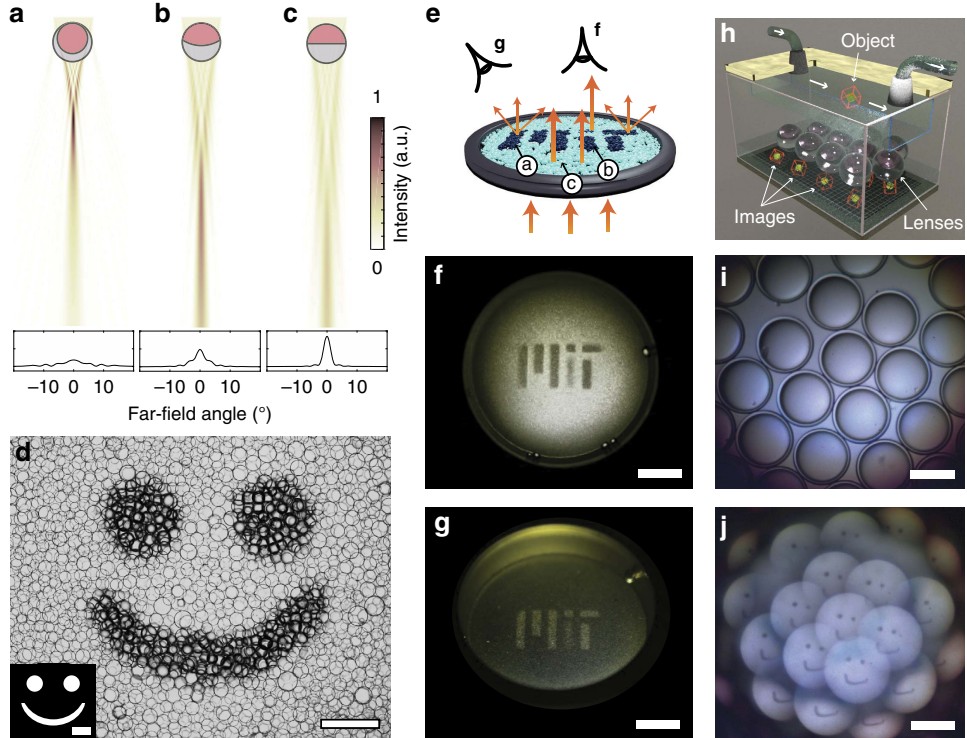

**Figure 4 | Towards potential applications.** (**a**–**c**) 2D finite difference time domain simulations of droplets of 5 µm radius for incident light of 500 nm wavelength. The internal radii of curvature are 4 µm (**a**), 9 µm (**b**) and infinite (**c**). (**d**) Localized exposure of light-sensitive surfactants to UV light leads to a local variation in droplet morphology and scattering behaviour, which is used to create an image. The dark zones in the image represent particles that have switched to the double emulsion state (**a**). They scatter light more strongly and therefore appear darker than the flat interface Janus droplets (**b**). The inset shows the photomask. Scale bars = 500 µm. (**e**) Schematic showing two different geometries for observing the photo-patterned droplet films, corresponding to the perceived images shown in **f**,**g**. The labels a, b and c correspond to the droplet geometries shown in panels **a**–**c**. (**f**,**g**) Photo-patterned droplets viewed from above (**f**) and at an angle (**g**). Scale bars = 5 mm (**f**,**g**). (**h**) Concept sketch for tomographic imaging of micro-scale objects in a microfluidic system using the fluid compound lenses. Each lens captures an image of the object (here a red cube with yellow sphere) at a different perspective. We anticipate that the three-dimensional shape of imaged objects can be reconstructed from the elemental images. (**i**) Monolayer array of fluid compound lenses. (**j**) Images projected by the monolayer lenses. (**i**,**j**) Scale bars = 100 µm.

resolution (for experimental details, see Methods). From the experimental PSF estimate we find that the droplets can resolve details down to a feature size of $r_{exp} = 3.7$ µm. The theoretical two-point resolution limit of a comparable diffraction limited lens is $r_{th} = 3.1$ µm. We also determined an estimate of the Modulation Transfer Function (MTF) of the same droplets based on the experimentally obtained PSF estimate. The MTF's cut-off frequency at a remaining image contrast of at least 10% of the original object contrast is found to be $f_{exp}^{10\%} = 0.22$ cycles per µm, which corresponds to a sinusoidal intensity variation of 4.5 µm period. The theoretical line pattern resolution limit for a comparable lens is 3.2 µm ($f_{th}^{10\%} = 0.31$ cycles per µm). The discrepancy between the measured and the theoretical resolution limits is likely due to spherical aberrations.

Using micro-fluidics we can produce emulsion lenses with a highly uniform size distribution[44]. This can be seen in Supplementary Fig. 7a. Droplets of the same size, with the same volume ratio and matching internal interface curvature have matching focal lengths. This is apparent in the images and data presented in Supplementary Fig. 7b–d, which show that when light is focused through several lenses of the same size, all PSFs have similar shapes. We therefore conclude that all lenses have very similar focal lengths, and numerical apertures.

**Potential applications of tunable droplet compound lenses.**
To explore potential application scenarios, we tested if the

micro-scale optical tunability of the droplets could be translated to observable differences in macroscopic properties. We hypothesized that the predicted variation of the far-field angular intensity distribution of light scattered by the lenses as a function of their internal curvature (Fig. 4a) would result in changes in the macroscopic appearance of droplet assemblies. In the case of a strongly focusing double emulsion, finite difference time domain (FDTD) simulations show that a single droplet will scatter light in a cone with an opening angle of almost 30°. On the other hand, the Janus droplet with a nearly flat interface transmits light with an angular spread of only a few degrees. To test whether this phenomenon could be observed to create droplet-based displays, we formed films of polydisperse emulsion droplets covering an area of several cm². The localized variation of the geometry of droplets in selected regions of the film was expected to lead to a visually perceivable change in its macroscopic appearance. In order to induce localized variations in droplet morphology, we employed an optically switchable azobenzene surfactant to change the morphology of the emulsions droplets, as previously described[44,48]. Irradiation of selected areas of droplets with UV light using a 'smiley face' photomask induces a transformation of the exposed droplets from the transparent Janus geometry to a strongly scattering double emulsion geometry. Simple visual inspection reveals a clear optical contrast when viewed in transmission (Fig. 4b). Exposure to UV radiation and blue light allows us to reversibly switch the compound lenses between these two morphologies

 

again and again, without any signs of degradation. This stability and reversibility is in good agreement with observations reported by other groups that investigated the dynamic performance of similar photo-active surfactants[49–52]. While we have not investigated the response time of the droplets quantitatively, we expect changes in lens geometry to occur on a time-scale of less than 1 s. In fact, azobenzene derivatives have been shown to switch configuration states on the timescale of milli- or even microseconds[53] and diffusion of the small surfactant molecules to and from the interface on the length scale of the droplet size occurs on millisecond timescales[54].

The FDTD simulations predicted that droplets with an internal curvature somewhere between the extremes of the double emulsion state and the flat-interface Janus configuration scatter light in a cone with an opening angle larger than that of the Janus droplets, but smaller than that of the double emulsions (Fig. 4a). We tested if these optical differences could create surfaces with controlled spatial variation in perceived brightness (Fig. 4c) by finely adjusting the droplet's internal curvature through careful tuning of the UV light exposure. To this end, we irradiated a droplet assembly through an MiT photomask in which a piece of scattering Scotch tape was placed over the stem of the 'i' to partially block UV transmission. Our expectation was that the partially blocked area would display smaller variations in curvature of the droplet-internal interfaces as compared to the fully exposed areas. Consistent with this design, we observed a significant decrease in pattern brightness in the modified photomask region of the sample when observed in direct transmission (Fig. 4d). The double emulsions scatter light into a larger angular range; consequently, when the same sample is viewed at an angle, the areas that were exposed to the UV radiation appeared brighter. Hence, we observe an inversion of the image (Fig. 4e) consistent with the FDTD simulations. In short, we can vary image contrast in the droplet films by photo-chemically modulating the degree of curvature of the droplets' internal interface.

The droplets' variable focal length and their capability to form images are properties that are particularly relevant for a variety of application scenarios related to miniaturized imaging devices. Arrays of micro-lenses, for example, find application in digital integral microscopic imaging and photography[1,3]. One of the main challenges in three-dimensional image acquisition is the limited depth of field[2]. The tunable focal length lenses could provide the means to address this limit. To evaluate whether the lenses could be considered for integral imaging applications, we produced monodisperse bi-phase double emulsion droplets and arranged them in a close-packed monolayer. This enables us to exploit the individual droplets' imaging capabilities in multi-droplet assemblies. In such a multi-lens arrangement, each lens projects a plane elemental image of an object at slightly different angles (Fig. 4f–h). Therefore, each lens has a different perspective of an imaged 3D object. Computational recombination of the images from multiple lenses should then allow for the capturing of the three-dimensional forms of imaged objects.

## Discussion

Complex emulsions of optically distinct, immiscible hydrocarbons and fluorocarbons in aqueous media can form droplets that act as compound lenses with a tunable droplet-internal optical interface. Adjustment of the droplet's interfacial tensions with the aqueous phase allows for a continuous and reversible variation from double emulsions, through Janus configurations, to inverted double emulsions. Depending on their configuration, the droplets show different interactions with light. Double

emulsions with the optically denser fluid as the droplet-core phase strongly focus light. Janus droplets do not significantly disturb the light wavefront, when the surface normal of the internal interface is aligned with the light propagation direction. Double emulsions with the optically denser fluid as the droplet-shell phase show strong light scattering. A controlled modification of the droplet morphology consequently results in a predictable variation of the droplets' light focusing and scattering behaviour.

We have shown that, depending on their morphology, the droplets can act as converging lenses projecting real inverted images, or as diverging lenses forming virtual upright images. These emulsion-based micro-lens droplets have a dynamically tunable focal length that can vary from $\pm 3.5 \times$ the drop diameter to $\pm$ infinity. With a resolution limit around $4\,\mu m$, the reconfigurable micro-lenses do not show diffraction-limited performance (resolution limit of $3\,\mu m$ for a comparable diffraction-limited micro-lens), which we attribute to the presence of spherical aberrations. While the resolution limit of standard high numerical aperture microscope objectives is at least an order of magnitude higher, the microscopic droplet compound lenses have clear advantages in applications where device size, simplicity and the ability to reconfigure on-demand matters; this could be of particular interest in synthetic aperture integral imaging where *in-situ* reconfigurable optical components could help to enhance performance of the imaging device[55]. Liquid lenses with variable focal length could form the basis of adaptive micro-scale optical elements in miniaturized integral 3D imaging and sensing devices.

We have also shown that such tunable droplet micro-lenses exhibit differences in their macro-scale optical appearance, which can be used for the creation of patterns and images. Light scattering is significantly more pronounced for droplets in the double emulsion geometry, while droplets with a flat internal interface induce much smaller perturbations in the propagating light wavefront. This allows for the creation of microscopic and macroscopic patterns with tunable contrast, which could form the basis for light field displays capable of creating 3D images and projecting variable information content into different directions.

For the present work, we rely on gravity to orient the lenses with their symmetry axis perpendicular to the substrate due to the density difference between the constituent lens components. Future investigations may address the control of lens orientation and droplet morphology with forces other than gravity, including using stimuli such as light, thermal gradients and electric fields. While we focused on heptane-FC-770 droplets, there are a number of other materials that could be used in order to adjust the refractive index of the droplet phases. We have also conducted experiments using lenses formed from hexane and perfluorohexane, which showed similar optical characteristics as the heptane-FC-770 droplets reported here. Using halogenated liquids with high refractive index as constituents of the emulsion droplets would enable the formation of compound lenses with higher refractive power. This additional degree of freedom—the choice of emulsion formulation—could be used for correcting aberrations or for introducing a desired chromaticity. Incorporation of active optical media, plasmonic elements or magnetic nanoparticles will open up a broad parameter space for tuning and controlling the fluid micro-lenses' dynamical optical properties, and simultaneously provide access to a multitude of enticing sensing paradigms and optical applications.

## Methods

**Droplet formation.** Double emulsion droplets were formed using a 1:1 volume ratio of heptane and Fluorinert FC-770 for the lensing experiments and

a 1:1 volume ratio of (2:1 hexane:heptane) to FC-770 for the UV switchable droplets. For each of the two material combinations, the two fluids were combined in equal volumes and heated to just above the suspension's upper critical solution temperature $T_c$ to allow the two liquids to form a homogeneous mixture. An aqueous surfactant solution heated above $T_c$ was then added, and the resulting mixture was quickly shaken to form small multi-disperse droplets, which were left to cool to allow the constituent oils to phase separate. Mono-disperse droplets were formed in a glass capillary microfluidic device that consists of an outer square capillary (outer diameter, 1.5 mm, inner diameter, 1.05 mm, AIT Glass), and an inner cylindrical capillary (outer diameter, 1 mm, World Precision Instruments). The capillary assembly was pulled to form a 30 μm tip using a P-1000 Micropipette Puller (Sutter Instrument Company). Harvard Apparatus PHD Ultra syringe pumps were used to inject the homogenous mixture of fluorocarbon and hydrocarbon into the inner capillary and aqueous surfactant solution into the outer capillary. The microfluidic device and syringe pumps were maintained at a temperature above $T_c$ using a heat lamp while the drops were formed, and the drops were then cooled below $T_c$ to induce phase separation. We closely followed the procedures reported previously[44]. The droplets were found to be stable on the timescale of several days, at least. We expect them to stay stable for much longer, provided the aqueous medium and sample environment are optimized (see Supplementary Note 5 and Supplementary Fig. 6).

**Determining the curvature of the internal interface.** The curvature of the internal interface between the droplets' hydrocarbon and fluorocarbon phases was determined with a custom-built microscope with horizontal axis that allowed capturing side views of the droplets. For these experiments, the droplets were placed onto a hydrogel substrate enclosed between two coverslips. The microscope consists of an Olympus ×5 objective (NA = 0.15), a Thorlabs tube lens (effective focal length = 200 mm) and an OMAX 14.0MP Digital USB Microscope camera. A white screen was placed behind the sample, and the sample was illuminated from the side using a Fiber-Lite MI-152 lamp.

When viewing the internal interface between the fluorocarbon phase and the hydrocarbon phase, the image is distorted due to the curved outer phase. This distortion due to refraction at the droplet's outer surface needs to be accounted for when determining the location and curvature of the droplet-internal interface. Therefore, we apply the following correction to find the position of an object—in this case a point on the internal interface—within a droplet of refractive index $n_1$ that is located in a medium with refractive index $n_2$: if the object is located at a distance $h$ measured perpendicular to the optical axis, which passes the centre of a sphere of radius $R$, the height of its image $h_i$ is given by the paraxial approximation (see Supplementary Note 3):

$$h_i = h \frac{n_1}{n_2}. \tag{3}$$

This implies that the actual height $h$ of an object in the droplet that *appears* to have a height $h_i$ will be:

$$h(h_i) = h_i \frac{n_2}{n_1}. \tag{4}$$

To deduce the actual droplet-internal interface location, this correction was applied by first determining the off axis distance $h_i$ for each point along the interface in the side view images of the drops using a custom MATLAB algorithm. We then used equation (4) to calculate the real shape of the internal interface by fitting a circle to this corrected curve to determine the interface curvature. The error associated with using the paraxial approximation is discussed in Supplementary Note 3 and Supplementary Figs 4 and 5.

**Focus scans.** A custom-build microscope was used to reconstruct the light field behind the lenses. For this experiment, the drops were illuminated with a quasi-monochromatic plane wave. This was achieved by imaging the output of an optical fibre with 50 μm core in the back focal plane of an NPL ×20 objective (Leitz Wetzlar, NA = 0.45) used as a condenser. A 540 nm bandpass filter with an 80 nm bandwidth was used to create quasi-monochromatic light. The light field behind the droplets was captured by scanning the focal plane in 5 μm steps using a Madcity Labs micro-stage, a ×20 Olympus objective (NA = 0.75), a 200 mm focal length achromatic doublet tube lens and an Andor Zyla sCMOS 5.5 Camera. The light field data was analysed using MATLAB and ImageJ. The location and size of the droplets were determined from the images using ImageJ's measurement tools. The data from the focus scans were then entered into MATLAB to reconstruct the 3D light field behind individual droplet lenses, similar to the approach previously used to measure the light field behind retinal cell nuclei[56]. After the light field was measured, the droplet lenses were placed in a microscope with horizontal optical axis and imaged from the side. This side view was used to determine the curvature and volume ratio of the drops.

**Measuring droplets' focal length.** In order to quantify the image formation characteristics of the droplets, an image of a grid pattern was projected in front of them using a ×60 Olympus water dipping objective (NA = 1.0). The droplets acting as lenses projected the object to form a new image, which was then recorded using a ×10 objective (NA = 0.3) with a customized microscope setup (Fig. 3a).

The distance of the input image $x_1$ to a droplet lens can be controllably varied, and the position of the projected image behind the lens $x_2$ is determined by locating the plane, where the projected image is in focus (Fig. 3b–d). The location of the input and recorded images are related to the focal length by the simple lens relation $(n_c/s_i) + (1/s_o) = (1/f)$ (see Supplementary Note 2 and Supplementary Fig. 2) where $s_o$ and $s_i$ are the distances from the input image to the first principle plane and from the second principle plane to the recorded image, respectively, and $n_c = (n_{m_1}/n_{m_2})$ is the refractive index contrast between the surrounding media before and after the droplet lens (in our measurement setup $m_1$ was water and $m_2$ was air). The focal length and principle plane locations ($p_1$ and $p_2$) could then be determined by measuring the location of the output image for various input image locations and by using the relation

$$\frac{n_{\text{water}}}{x_1 - p_1} + \frac{1}{x_2 - p_2} = \frac{1}{f}, \tag{5}$$

where $x_1 - p_1 = s_i$, and $x_2 - p_2 = s_o$.

**Determining the droplets' PSF and MTF.** The droplet lenses' PSF and MTF provide quantitative measures of the two-point resolution and line pattern contrast limits that can be achieved when the lenses are employed in imaging applications. To get an estimate of the PSF and MTF we expose individual droplets to white light, which originates from an optical fibre with a 50 μm diameter core and is collimated by a spherical lens ($f = 150$ mm). We capture the droplet's PSF using a custom-build microscope composed of a ×50 Olympus objective (NA = 0.5), a Thorlabs tube lens ($f = 200$ mm) and an Allied Vision ProSilica GT3300C camera. We only use the camera's red channel (maximum quantum efficiency at 620 nm), expecting that the droplets' resolution would be at least comparable or better for smaller wavelengths. We obtain the MTF by Fourier-transforming the PSF after subtraction of background noise (which is legitimate because we do not expect diffuse scattering from the clear droplet), removal of 'salt and pepper' noise (due to hot pixels) using a median filter in a 3 × 3 pixel neighbourhood area, and averaging over angular slices of the imaged Airy disk pattern. The resulting PSF and MTF curve must be seen as a conservative estimate as the recorded PSF still possesses residual blur from the microscope optics.

**Using UV light to switch lens morphology.** With a light-sensitive surfactant containing an azobenzene moiety, 3-(4-((4-butylphenyl)diazenyl)phenoxy)-N,N,N-trimethylpropan-1-aminium bromide, in the aqueous medium, the fluid lenses can be switched from a transparent Janus state to a scattering double emulsion state, and back, simply by exposure to light in the UV and blue spectral ranges as described previously[48]. The droplets consisted of a 1:1 volume ratio of (2:1 hexane:heptane) to FC-770 and a 100 μl total volume was used. The outer phase was composed of 600 μl of 0.1 wt% azobenzene surfactant and 80 μl of 2 wt% Zonyl FS-300 in water. A laser-printed photomask transparency displaying a smiley or the MIT logo was placed beneath the droplets in a dish on the stage of an inverted microscope. In the case of the MIT logo, a piece of semi-transparent Scotch table was placed over the 'i' to partially block light transmission to the sample and to induce a grey-scaling effect. The sample of liquid lenses, initially in a Janus morphology, was then illuminated with UV light through the photomask (DAPI filter, $\lambda = 365$ nm) to induce transformation of the droplets in the exposed areas to the double emulsion state. This light-induced reconfiguration of the lens-internal interface can be reversed by exposure to blue light (through a FITC filter, $\lambda = 470 \pm 20$ nm). Gelatin can be added to the outer aqueous phase to reduce the rate of diffusion and prolong image persistency.

**Optical simulations.** In all simulations, the overall droplet shape was assumed to be spherical, which is a valid assumption provided that the interfacial tensions $\gamma_{\text{FH}}$ between fluorocarbon and hydrocarbon, $\gamma_F$ between fluorocarbon and the aqueous medium, and $\gamma_H$ between hydrocarbon and the aqueous medium, satisfy the relations $\gamma_{\text{FH}} \ll \gamma_F \approx \gamma_H$ (ref. 44). The internal interfaces were assumed to be spherical, based on the following argument: interfaces between liquids can be considered to be spherical when the ratio of gravitational forces to surface tension forces is small. This ratio is given by the Bond number $Bo = ((\Delta\rho \cdot g \cdot L^2)/\gamma_{\text{FH}})$, where $\Delta\rho$ is the difference in density of the two droplet phases, $g$ the gravitational constant and $L$ the droplet diameter[57]. For a material system similar to the one used here, such as the hexane-perfluorohexane bi-phase droplets with a diameter of 100 μm, constituent densities of $\rho_{\text{HX}} = 0.66$ g cm$^{-3}$ and $\rho_{\text{FHX}} = 1.68$ g cm$^{-3}$ and a surface tension $\gamma_{\text{FH}} = 0.4$ mN m$^{-1}$ (ref. 57), the Bond number is around 0.25. While a Bond number of around 0.1 is usually considered to be an upper limit for assuming spherical curvature of a liquid–liquid interface[58], images of our complex droplets show that the interface of the droplets obtained can be approximated reasonably well with a spherical fit (inset in Fig. 3e). Deviations are apparent closer to the triple-phase contact line, but these regions do not strongly affect the optical behaviour of the droplet lenses.

FDTD simulations were completed using the open source software package MIT Electromagnetic Equation Propagation[59]. Double emulsions and Janus droplets of Heptane ($n_H = 1.387$) and FC-770 ($n_F = 1.27$) with a radius of 5 μm in water ($n_W = 1.33$) were illuminated with a 500 nm wavelength monochromatic line light source. Equal volumes of Heptane and FC-770 were simulated, and the overall

droplet shape and the shape of the interface were assumed to be spherical, such that equation (1) yields the interface location. A perfectly matched layer was used as the boundary condition on all edges of the cell, and the resolution was 32 units per μm. The simulation was run until a steady state was reached where the intensity no longer varied between time steps.

Ray-tracing was implemented in MATLAB. Each ray was defined by its location, direction, intensity and polarization. The rays were propagated to the drop and were refracted and reflected at each interface of the drop. The direction vector $\vec{d}_t$ of the refracted ray was determined using a vector version of Snell's law:

$$\vec{d}_t = \frac{n_1}{n_2}\vec{d}_i + \left( \frac{n_1}{n_2}\cos(\theta_i) - \sqrt{1 - \left(\frac{n_1}{n_2}\right)^2 \left[1 - \cos^2(\theta_i)\right]} \right)\vec{n}, \qquad (6)$$

where $\vec{d}_i$ is the direction of the incident ray, $\vec{n}$ is the surface normal, $\theta_i$ is the angle that the incoming ray makes with the surface normal, and $n_1$ and $n_2$ are the refractive indices before and after the interface (see Supplementary Note 4). The intensity of the refracted and reflected rays was calculated using the Fresnel Equations[45].

The ray transfer matrix was calculated numerically using MATLAB. The transfer matrix consisted of the product of transfer matrixes for a ray entering the drop (water to heptane), propagating a distance $R_d + l$ to the interface between heptane and FC-770, being refracted at this interface, propagating the rest of the way through the drop, and being refracted at the outer interface of the drop (FC-770 to water). In order to compare with experiments, refraction through the coverslip after the drop was included in the ray transfer matrix. The locations of the focal points were consistent between the ray transfer matrix and ray-tracing.

**Data availability.** The data that support the findings of this study are available from the corresponding author on request.

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

## Acknowledgements

We thank Frank Grunert for introducing us to the dynamic mathematics software GeoGebra. S.N. and M.K. gratefully acknowledge support by the National Science Foundation through the 'Designing Materials to Revolutionize and Engineer our Future' programme (DMREF-1533985). S.N. acknowledges support by NSERC through a Graduate Fellowship. T.M.S. and L.D.Z. acknowledge support of the National Science Foundation through grant DMR-1410718. Mo.K. and K.S. were supported by the Max Planck Society. M.K. thanks the MIT Department of Mechanical Engineering for support.

## Author contributions

L.D.Z., S.N., T.M.S. and M.K. conceived the project. T.M.S., Mo.K., D.B. and M.K. supervised and supported the project. S.N. performed the optical and morphological modelling, for which V.S, D.B. and G.B provided guidance. L.D.Z., S.N. and N.N. fabricated the emulsion lenses. J.A.K. synthesized the light-sensitive surfactants. K.S., Mo.K. and S.N. implemented the optical characterization setups. S.N., K.S. and L.D.Z. performed the optical experiments, for which Mo.K., G.B. and M.K. provided guidance. S.N, L.D.Z. and M.K. analysed the data. All authors discussed the results and contributed to the writing of the manuscript.

## Additional information

**Competing financial interests:** The authors declare no competing financial interests.

**Publisher's note**: 

