## [Peer Review File · Nature Communications]

Reviewers' comments:

Reviewer #1 (Remarks to the Author):

In this paper entitled "Reconfigurable and responsive droplet-based compound micro-lenses", the authors present a new generation of fluidic tunable compound micro-lenses. The authors systematically characterized the droplet lenses' optical characteristics such as capturing imaging, PSF, MTF, and also demonstrated some potential applications of the micro-lenses. Although the liquid lenses have been extensively researched over the past decade, the successful demonstration of reversible switching of focus length through using emulsion droplets triggered by the external light is new. However, at the current stage, the paper is not very clearly written and the following major comments should be fully addressed before the recommendation of the paper for publication.

1: The last image in Fig.1d is wrongly presented. The core of the droplet should be gray instead of red.

2. One of the key points of this work is to use the surfactant to adjust the focus length by tailoring the liquid-liquid interfacial tension. Thus, it is important to show the stability of the surfactant in the aqueous environment, especially when exposed to the laser light. Moreover, in the case of the dynamic focus length switching using the laser light, the response time of the compound droplet should be characterized?

3. Another important concern is the stability of the refractive index of the droplet lenses. In particular, both the hexane and heptane used in this work are volatile, and the evaporation of both liquids is susceptible to the instability of their refractive index over time. In this connection, more detailed discussion should be provided.

4. The focus length is sensitive to the internal curvature of droplet. For the double emulsion droplet with the identical core size, how the variation of the shell thickness of the imaging influences the experimental results such as focal length, imaging quality.

Reviewer #2 (Remarks to the Author):

Review for NCOMMS-16-20346-T "Reconfigurable and Responsive Droplet-Based Compound Micro-Lenses" by Sara Nagelberg et. al.

The paper presents a new type of compound micro-lenses with the possibility to dynamically change their focal length. The lenslet are built of bi-phase emulsion droplets, following the concept introduced in Ref. 44. The microfluidic lenses that can tune the $f/\#$ in the range of ± 3.5 to ∞ are demonstrated. The optical properties of the lenses are analyzed theoretically, by simulation, and experimentally.

Adjustable reconfigurable microlenses may find application in various optical sensing and display techniques. The paper demonstrates a droplet based display and show an initial experiment demonstrating the applicability for 3D integral imaging. Building integral imaging systems with micro-adjustable lenses is indeed an old dream. Using micro-adjustable lenses can be used to overcome the inherent space -bandwidth (information throughput) limitation of the technique. Recently liquid crystal lenses and electrowetting lenses were proposed in integral imaging , however none have been realized at microscale . The microfluidic compound lenslets may provide a new powerful tool for the engineering of 3D integral imaging sensing and display systems. Particularly it can be used for integral

microscopy, (aka "light field microscopy", "plenoptic microscopy", "iMic") to extend the depth of field and the resolution. Such devices can be used for 3D displays with adjustable floating image region ("central depth planes"). It can be very useful for the design of switchable 2D/3D displays. Other fields that may benefit of the droplet microfluidic lenses are: optical biosensing, waveform and beam shaping, light harvesting, for optical tweezers.

One of the limitations of the presented droplet lenses is that their optical axis needs to be aligned with the gravitation field. This isn't an issue for integral microscopy, but it can limit many of the other applications. The authors acknowledge this problem in the Outlook and propose techniques to overcome it.

Comments:

1. In Fig. 2 b please specify where is the distance measured from? The reader may guess that it is measured from the droplet, which apparently isn't so (it wouldn't match the NA values in Fig. 3)
2. In Fig. 3g the area of the experimental PSF appears to be larger than the area of the theoretical PSF. Is it because differences in the side lobes? Nonlinear processing of the data (e.g. the median 3x3 filter) ? The PSFs should be normalized. Such normalization shouldn't affect the zero crossing or the Rayleigh resolution.
3. The NA value of 1.2 reported on Page 5 line 201 and in the captions of Figure 3 is much higher than the maximum achievable NA of ~ 0.15 (from the reported $f/\# = 3.5$ and from Fig. 3f)
4. What do the "f" and "g" labels in Fig. 4c represent?
5. Why were the FDTD simulations run on droplets with a radius of $5\mu\text{m}$ (page 11, line 454) when most of the other demonstrations are with droplets with a radius of $\sim 50\mu\text{m}$?
6. Could the authors comment on the uniformity of the droplets? How uniform is their focal length?
7. What is the time response to the UV illumination? How fast is it possible to change the microlenses internal shape /focal length? How precise? Do they exhibit any hysteresis?
8. In Eq. (S5) in the supplement, the terms n_1 and n_2 should switch sides of the equation. Yet the consequent equations are correct.
9. In Eq. (S13) in the supplement, there should be a negative "-" sign instead of "+". Yet the consequent equations are correct.

Reviewer #3 (Remarks to the Author):

This manuscript reports on focal-tunable microlenses fabricated using bi-phase emulsion droplets. By adjusting the droplets' interfacial tensions with the aqueous phase, the curvature of the interface between hydrocarbon and fluorocarbon liquids can be changed correspondingly, resulting in a shift of the focal point of the droplet-based microlens. The authors studied focusing and imaging capabilities of the emulsion lenses by FDTD simulations and experiments, and also showed potential applications of the lenses, such as 3D light field display and 3D imaging.

The presented work is quite complete. However, this reviewer has a few major concerns. The first is the novelty of the general approach taken. Very similar results were published before (Nature. 2015 Feb 26; 518(7540): 520–524), including materials and fabrication process. Even the structure of the lenses in this manuscript is the same as the previous work. Although the authors show some new work by claiming a tunable liquid lenses as well as their optical characteristics and simulations, the core technology still lies in the emulsion droplets, hence lacking the necessary novelty for publication in Nat. Comm. Second, although the bi-phase emulsion microlens presented here shows some advantages, such as slight increase in the focal tunable range, it would not be considered a quantum leap compared to other tunable liquid lenses that have been widely reported. The tunable liquid lens is

not a new concept, and many other approaches have been applied in previous works. What is presented here in terms of the lenses is more leaning towards an alternative approach, rather than a significant breakthrough, be it optical performance or responsive time of the lenses.

Minor issues:

- 1) By microfluidics assisted assembly of liquid lenses, the droplets are confined by glass channels, which may cause slight deformation of lenses due to the pressure from the sidewalls or bottom of the channel. Did authors consider the influences of the droplet deformation on the optical performance of the microlenses?
- 2) During the formation of the bio-phase emulsion droplets, is it possible to control the alignment of the two liquids so that the direction of the optical axis of the microlenses can be precisely controlled?
- 3) What's the responsive time of the microlenses in this work, for both thermal and UV exposure stimulus?

Overall, this reviewer finds this work more appropriate for a special topic journal than a high-impact one such as Nat. Comm.

REVIEWERS' COMMENTS:

Reviewer #1 (Remarks to the Author):

The authors have fully addressed my previous comments on the stability of the surfactant and corrected typos in their figures. I am supportive of the publication of the interesting work. Only very minor comment: The scale bar in Fig. 3b c and Fig. 4fg should be mentioned.

Reviewer #2 (Remarks to the Author):

The authors have made the appropriate corrections and satisfactorily addressed my concerns. I recommend the publication of this manuscript. The paper offers a new type of miniature liquid microlens array technology with a tuning mechanism that differs from the traditional ones (that are commonly mechanical or electrical driven). Therefore it provides a new and powerful approach with a potential impact in all the fields and application targeted in the quite extensive microfluidic lens research.

One can think on a broad spectrum of applications that may benefit of the presented technology. Besides the light filed and 3D displays considered in the paper in the paper, the low cost and fast tunable microlenses can be of interest in many applications such as: bio-sensing, ophthalmology, endoscopy, optical harvesting, computational optical sensing, illumination shaping, photolithography, optical trapping , and more.

Response to Reviewers' Comments on Manuscript NCOMMS-16-20346-T

“Reconfigurable and Responsive Droplet-Based Compound Micro-Lenses”

by Sara Nagelberg, et al.

Reviewer #1

In this paper entitled “Reconfigurable and responsive droplet-based compound micro-lenses”, the authors present a new generation of fluidic tunable compound micro-lenses. The authors systematically characterized the droplet lenses’ optical characteristics such as capturing imaging, PSF, MTF, and also demonstrated some potential applications of the micro-lenses. Although the liquid lenses have been extensively researched over the past decade, the successful demonstration of reversible switching of focus length through using emulsion droplets triggered by the external light is new. However, at the current stage, the paper is not very clearly written and the following major comments should be fully addressed before the recommendation of the paper for publication.

1: The last image in Fig.1d is wrongly presented. The core of the droplet should be gray instead of red.

We thank the reviewer for pointing out this inconsistency in Fig. 1, which we have corrected in the revised manuscript.

2. One of the key points of this work is to use the surfactant to adjust the focus length by tailoring the liquid-liquid interfacial tension. Thus, it is important to show the stability of the surfactant in the aqueous environment, especially when exposed to the laser light. Moreover, in the case of the dynamic focus length switching using the laser light, the response time of the compound droplet should be characterized?

Emulsion droplets in our experiments and in the work reported in reference 44 proved to be stable for a period of several days. This stability relies on conditioning the aqueous medium hosting the droplets and keeping the system isolated from the ambient environment to prevent diffusion of the droplets’ constituent fluids. This is discussed in more detail in the answer to the reviewer’s next comment.

Using exposure to UV light and blue light allows us to reversibly switch the compound lenses between different morphologies again and again, without any signs of degradation. The observed stability and reversibility is in good agreement with observations reported by other groups that investigated the dynamic performance of photo-active surfactants with similar substitution patterns (para-substituted, alkyl and donor substituents):

- Kienzler M. A., Reiner, A., Trautman, E., Yoo, S., Trauner, D., and Isacoff, E. Y., A Red-Shifted, Fast-Relaxing Azobenzene Photoswitch for Visible Light Control of an Ionotropic Glutamate Receptor. *J. Am. Chem. Soc.* **2013** *135* (47), 17683-17686.
⇒ Over 600 switching cycles are reported in this paper (see graphics in supporting info).
- Mahimwalla, Z., Yager, K.G., Mamiya, J. et al. Azobenzene photomechanics: prospects and potential applications. *Polym. Bull.* (2012) *69*: 967.
⇒ A review that claims 10^5 - 10^6 cycles.
- Klajn, R., Wesson, Paul J., Bishop, Kyle J. M. and Grzybowski, Bartosz A. (2009), Writing Self-Erasing Images using Metastable Nanoparticle “Inks”. *Angew. Chem. Int. Ed.*, *48*: 7035–7039. doi:10.1002/anie.200901119.
⇒ At least 300 cycles are claimed in this report.

- Shang T, Smith K. A., Hatton T. A., Photoresponsive Surfactants Exhibiting Unusually Large, Reversible Surface Tension Changes under Varying Illumination Conditions. *Langmuir* **2003** 19 (26), 10764-10773.

The time that the droplets take to switch between different configurations upon exposure to UV / blue light relies on a trans-to-cis conformational change in the azobenzene section of the surfactant molecules and the time required for the molecules to dissociate or associate with the droplet interface. Azobenzene derivatives have been shown to switch configuration states on the timescale of milli- or even microseconds [García-Amorós, J. & Velasco, D. Recent advances towards azobenzene-based light-driven real-time information-transmitting materials. *Beilstein Journal of Organic Chemistry* **8**, 1003–1017 (2012)]. This translates into a reconfiguration of the compound lens geometry on a timescale of the order of 1s or less. A real-time video of the light-induced lens reconfiguration can be found here:

www.nature.com/nature/journal/v518/n7540/fig_tab/nature14168_SV4.html.

3. Another important concern is the stability of the refractive index of the droplet lenses. In particular, both the hexane and heptane used in this work are volatile, and the evaporation of both liquids is susceptible to the instability of their refractive index over time. In this connection, more detailed discussion should be provided.

The solubility of hexane, heptane, and FC770 is extremely low in water; however, over sufficiently long time scales, diffusion of hexane, or heptane through the aqueous medium into the ambient environment can lead to changes in droplet morphology. This is easily prevented by keeping the droplets and the medium in a closed environment and by suppressing diffusion into the aqueous medium, which can be achieved through priming with the respective solvent. FC770, a long-chain fluorinated oil, was not found to diffuse into the aqueous medium on the timescale of days. Following the reviewers comment, we have added a figure below (Fig. R1), which visualizes slow droplet morphology variation in suboptimal experiment conditions (Fig. R1a). Droplet morphologies are stable, if the experimental conditions are appropriately controlled (Fig. R1b). This figure was also included in the supplementary information as Figure S6. For the droplets consisting of heptane and FC770, we saturate the aqueous medium with heptane, to avoid heptane diffusion from the droplets into the medium. Longer-term stability tests have not yet been conducted but will be addressed in future work. For the stabilized droplet morphologies, we expect no change in refractive index over time.

Figure R1: **Droplet stability.** **a)** Variation of droplet morphology composed of heptane and FC770 over time, if diffusion of heptane is not suppressed. **b)** Droplet morphologies are stable, if diffusion of heptane is suppressed by enclosing the system and priming the aqueous medium with heptane. Scale bars: 100 μ m.

4. The focus length is sensitive to the internal curvature of droplet. For the double emulsion droplet with the identical core size, how the variation of the shell thickness of the imaging influences the experimental results such as focal length, imaging quality.

Inspired by the reviewer's question we have performed additional theoretical modeling to determine the influence of a variation in the thickness of the outer shell. The resulting data is shown in Fig. R2. The distance of focus decreases with increasing shell thickness. In the simulations we have only considered rays that enter both phases. Rays that only interact with the outside phase, which has a lower refractive index than the surrounding medium, diverge.

Figure R2: **Focusing distance as a function of droplet outer radius** R_d for fixed inner radius R_i . **a)** Raytracing diagrams for four different ratios R_d/R_i . **b)** Focus distance L measured from the point of light entrance into the droplet in units of R_i as a function of the ratio R_d/R_i .

Another interesting aspect related to the reviewer's question is the variation of focal length with volume ratio. We explored this aspect using raytracing for a constant overall droplet size and different fixed contact angles at the triple phase line. The resulting data is shown in Fig. R3.

This data reveals that a variation of the volume ratio of hydro- and fluorocarbon droplet constituents at constant triple phase line contact angles causes a variation in radius of curvature, ultimately leading to a predictable change in effective focal length. The information shown in Figure R3 has also been added to the supplementary information as Figure S3.

Figure R3: **Variation of focal length as a function of volume ratio**. **a)** Ray tracing diagrams for droplets with volume ratios of hydro- to fluorocarbon $V_H/V_F = 0.5, 1,$ and 2 . Here, the contact angle measured between the fluorocarbon phase and the aqueous medium at the triple phase contact line is kept constant at $\theta_F = \pi/4$. **b)** Effective focal length in units of droplet radius f_{eff}/R_d plotted against volume ratio V_H/V_F , for contact angles θ_F of $0.1, \pi/6,$ and $\pi/4$.

Reviewer #2

The paper presents a new type of compound micro-lenses with the possibility to dynamically change their focal length. The lenslets are built of bi-phase emulsion droplets, following the concept introduced in Ref. 44. The microfluidic lenses that can tune the $f/\#$ in the range of ± 3.5 to ∞ are demonstrated. The optical properties of the lenses are analyzed theoretically, by simulation, and experimentally.

Adjustable reconfigurable microlenses may find application in various optical sensing and display techniques. The paper demonstrates a droplet based display and show an initial experiment demonstrating the applicability for 3D integral imaging. Building integral imaging systems with micro-adjustable lenses is indeed an old dream. Using micro-adjustable lenses can be used to overcome the inherent space –bandwidth (information throughput) limitation of the technique. Recently liquid crystal lenses and electrowetting lenses were proposed in integral imaging, however none have been realized at microscale. The microfluidic compound lenslets may provide a new powerful tool for the engineering of 3D integral imaging sensing and display systems. Particularly it can be used for integral microscopy, (aka "light filed microscopy", "plenoptic microscopy", "iMic") to extend the depth of field and the resolution. Such devices can be used for 3D displays with adjustable floating image region ("central depth planes"). It can be very useful for the design of switchable 2D/3D displays. Other fields that may benefit of the droplet microfluidic lenses are: optical biosensing, waveform and beam shaping, light harvesting, for optical tweezers.

One of the limitations of the presented droplets lenses is that their optical axis needs to be aligned with the gravitation field. This isn't an issue for integral microscopy, but it can limit many of the other applications. The authors acknowledge this problem in the Outlook and propose techniques to overcome it.

Comments:

1. In Fig. 2 b please specify where is the distance measured from? The reader may guess that it is measured from the droplet, which apparently isn't so (it wouldn't match the NA in Fig. 3)

The distance recorded in Fig. 2b in the manuscript was measured from the plane at the very bottom of the droplet. This plane does not represent the droplet's principal plane, which we expect to be located slightly below the droplet. We show this experimental data to convey that the droplets indeed show a variation in their focusing power as a function of the internal curvature. However, we have refrained from a more comprehensive analysis of this particular dataset for the following reason: for this set of experiments, the volume ratio was not controlled to the same extend as was done for all other experiments that form the basis of the data in Figure 3 and 4. In fact, the volume of heptane was lower than the volume of FC770 for the droplets shown in Figure 2.

We politely disagree with the reviewer regarding the NA: For instance for the left image in the manuscript's Figure 2 with the shortest focal length we find an $NA \leq 0.14$. We determine NA here, using the divergence angle of light determined from the image at the focus, which amounts to $\alpha \leq 8^\circ$ (see Fig. R4). Here, the internal droplet phase, especially for droplets with a volume ratio $v_r = V_{\text{Heptane}}/V_{\text{FC770}} < 1$, contributes primarily to the focussing of the light, since rays that only traverse the outer phase are diverging.

Figure R4: Section of Fig. 2 with divergence angle at focus a marked.

Determination of the NA based on the droplet radius R_i gives a value that matches the one calculated above.

2. In Fig. 3g the area of the experimental PSF appears to be larger than the area of the theoretical PSF. Is it because differences in the side lobes? Nonlinear processing of the data (e.g. the median 3x3 filter) ? The PSFs should be normalized. Such normalization shouldn't affect the zero crossing or the Rayleigh resolution.

We thank the reviewer for pointing out that the area under the point spread function (PSF) should be normalized instead of the maximum of the point spread function as we originally had done in the initial manuscript. The normalization of the area of the PSF results from the commonly applied practice to normalize the zero-frequency value of the Modulation Transfer Function (MTF), i.e. $MTF(0) = 1$. This is a valid choice, since usually the contrast relative to the total amount of detected light is of primary interest. Since the PSF and MTF are a Fourier transform pair, we obtain $MTF(0) = \int_{-\infty}^{\infty} PSF(r)e^{i2\pi r \cdot 0} dr = \int_{-\infty}^{\infty} PSF(r)dr$, where r is the distance from the center of the cylindrically symmetric PSF. That is, if we follow the convention of $MTF(0) = 1$, then the area under the PSF has to be normalized as the reviewer correctly suggested. Figure 3g has been updated accordingly.

3. The NA value of 1.2 reported on Page 5 line 201 and in the captions of Figure 3 is much higher than the maximum achievable NA of ~ 0.15 (from the reported $f/\# = 3.5$ and from Fig. 3f)

We thank the reviewer for pointing out this error. The NA of the droplets is 0.12. We corrected this in the manuscript.

4. What do the "f" and "g" labels in Fig. 4c represent?

Thank you for catching that! We fixed the labels in Fig. 4c to refer to d, and e; they referenced the wrong figure parts in the initial manuscript.

5. Why was the FDTD simulations run on droplets with a radius of $5\mu\text{m}$ (page 11, line 454) when most of other the demonstrations are with droplets with a radius of with $\sim 50\mu\text{m}$?

Our procedures allow us to form droplets with less than $10\mu\text{m}$ diameter. We decided to do the FDTD simulations for the limit of small droplets to determine if their behavior can still be explained within the geometrical optics approximation without considering the role of diffraction effects. This is relevant in the context of the presented prototype micro-droplet lens display with a highly multi-disperse lens size distribution (see Fig. 4b in the manuscript). Comparison with FDTD simulations performed with $50\mu\text{m}$ lenses (see Fig. R5) show that the general observations hold true, although diffraction plays a stronger role for the smaller lenses, as expected.

6. Could the authors comment on the droplet uniformity? How uniform is their focal length?

Using micro-fluidics we can produce emulsion lenses with a highly uniform size distribution. This can be seen in Fig. S6

Figure R5: **FDTD simulations for $50\mu\text{m}$ sized lenses** with different internal morphology matching the $5\mu\text{m}$ lenses shown in Fig. 4a in the manuscript.

(attached below), Figure 4g of this manuscript, and in Figures 1b, 2b of reference 44. Droplets of the same size, with the same volume ratio and matching internal interface curvature have matching focal lengths. This is apparent in the images and data presented in Figure R6, which shows that when light is focused through several lenses of the same size, all point spread

Figure S6: **Droplet homogeneity and point spread function uniformity.** **a)** An array of uniform droplet lenses. **b)** Point spread functions (PSF) of the droplets shown in (a). **c)** Fit to the Airy disk using the Bessel function of the first kind of order one. **d)** Distribution of the minimum in the Airy pattern, which is a measure of the Rayleigh two-point resolution limit.

functions have similar shapes. This lets us conclude that all lenses have very similar focal lengths. This data has been added to the supplementary information as Figure S7.

7. What is the time response to the UV illumination? How fast is it possible to change the micro-lenses internal shape / focal length? How precise? Do they exhibit any hysteresis?

Using exposure to UV light and blue light allows us to reversibly switch the compound lenses between different morphologies again and again, without any signs of degradation. We provide a detailed discussion of the time response to UV illumination in the answer to the second comment of reviewer #1. We have not observed any hysteresis in the contexts of our experiments.

The precision of switching depends on the stimulus, and more specifically, on how well the stimulus can be controlled. Precise switching of the droplet morphology relies on precise control of the contact angle at the triple-phase contact line. This in turn is based on precisely adjusting the concentration of fluorocarbon and hydrocarbon surfactant in the medium for chemical actuation, or good control of illumination intensity and duration if the droplets are switched

using light and light-sensitive surfactants. In summary, precision will primarily depend on how well the stimulus that triggers droplet reconfiguration can be controlled. This question should be the subject of more detailed future investigations, in the context of specific application scenarios.

8. In Eq. (S5) in the supplement, the terms n_1 and n_2 should switch sides of the equation. Yet the consequent equations are correct.

The reviewer is correct in pointing out this mistake, which we have fixed. We thank the reviewer for their very careful evaluation of our paper and for pointing out this mix-up of the refractive indexes and the missing “-“ sign below!

9. In Eq. (S13) in the supplement, there should be a negative “-“ sign instead of “+“. Yet the consequent equations are correct.

This mistake is now corrected. Thank you for pointing it out!

Reviewer #3

This manuscript reports on focal-tunable microlenses fabricated using bi-phase emulsion droplets. By adjusting the droplets’ interfacial tensions with the aqueous phase, the curvature of the interface between hydrocarbon and fluorocarbon liquids can be changed correspondingly, resulting in a shift of the focal point of the droplet-based microlens. The authors studied focusing and imaging capabilities of the emulsion lenses by FDTD simulations and experiments, and also showed potential applications of the lenses, such as 3D light field display and 3D imaging.

The presented work is quite complete. However, this reviewer has a few major concerns. The first is the novelty of the general approach taken. Very similar results were published before (Nature. 2015 Feb 26; 518(7540): 520–524), including materials and fabrication process. Even the structure of the lenses in this manuscript is the same as the previous work. Although the authors show some new work by claiming a tunable liquid lenses as well as their optical characteristics and simulations, the core technology still lies in the emulsion droplets, hence lacking the necessary novelty for publication in Nat. Comm. Second, although the bi-phase emulsion microlens presented here shows some advantages, such as slight increase in the focal tunable range, it would not be considered a quantum leap compared to other tunable liquid lenses that have been widely reported. The tunable liquid lens is not a new concept, and many other approaches have been applied in previous works. What is presented here in terms of the lenses is more leaning towards an alternative approach, rather than a significant breakthrough, be it optical performance or responsive time of the lenses.

Minor issues:

1) By microfluidics assisted assembly of liquid lenses, the droplets are confined by glass channels, which may cause slight deformation of lenses due to the pressure from the sidewalls or bottom of the channel. Did authors consider the influences of the droplet deformation on the optical performance of the microlenses?

At every point in our experiments, we have taken great care to ensure that we only characterized droplets that were not in contact with the sample cell walls. Droplets that are in contact with the sample cell glass side walls are easily distinguishable and we have taken care to not include them in our analysis.

2) *During the formation of the bio-phase emulsion droplets, is it possible to control the alignment of the two liquids so that the direction of the optical axis of the microlenses can be precisely controlled?*

The difference in density of the distinct fluids forming the droplets results in an alignment of their axis of symmetry with the direction of gravity. If formed in a standard lab environment, the droplets will therefore always be aligned with their optical axis vertically. In all experiments we have taken this into account and designed the characterization set-ups for imaging through the droplets and focal length determination accordingly. In ongoing work we are exploring the alignment of the droplets using thermal gradients, electric, and magnetic fields, which will be discussed in a follow-up publication.

3) *What's the responsive time of the microlenses in this work, for both thermal and UV exposure stimulus?*

The response time of the droplets when exposed to UV / blue light to switch between different droplet configurations relies on a trans-to-cis conformational change in the azobenzene section of the surfactant molecules. Azobenzene derivatives have been shown to switch configuration states on the timescale of milli- or even microseconds [García-Amorós, J. & Velasco, D. Recent advances towards azobenzene-based light-driven real-time information-transmitting materials. *Beilstein Journal of Organic Chemistry* **8**, 1003–1017 (2012)].

This translates into a reconfiguration of the compound lens geometry on a timescale of order of 1s or less. A real-time video of the light-induced lens reconfiguration can be found here:

www.nature.com/nature/journal/v518/n7540/fig_tab/nature14168_SV4.html. Exposure to higher light levels will likely result in faster switching times.

We do not report on thermal stimulation of the lenses in this paper.

Overall, this reviewer finds this work more appropriate for a special topic journal than a high-impact one such as Nat. Comm.

We appreciate the reviewer's opinion but would like to state here that we politely disagree with their assessment. We feel that our work is appropriate for *Nature Communications*: it is highly multidisciplinary and will be of significant interest to the broad scientific community and especially to those conducting research in the fields of optics, optical device design, materials science and engineering, chemistry, medical imaging, physics, surfaces and interfaces, responsive and reconfigurable materials, emulsions, and complex fluids. Materials and devices displaying dynamic behavior have become of general interest to our society. This is apparent in the rapidly increasing demand for “smart” materials and systems allowing for multiple functions with superior performance, environmental adaption, and energy efficiency. The results presented in this manuscript have clear implications for the engineering of novel optical devices based on exploiting complex liquids in the form of droplets with varying optical properties. Our work forms the basis for the design of novel miniaturized imaging devices, 3D light field systems, solar energy collection enhancers, and bio-sensors — all applications of direct and accessible relevance to the general public. Furthermore, the new concepts of tuning the optical properties of micro-lenses presented in the manuscript are highly visual and readily apparent, allowing readers without significant background in optics and materials science to appreciate the results and implications of our research.